# Air Quality of Work, Residential, and Traffic Areas during the COVID-19 Lockdown with Insights to Improve Air Quality

**DOI:** 10.3390/ijerph19020727

**Published:** 2022-01-10

**Authors:** Badr H. Alharbi, Hatem A. Alhazmi, Zaid M. Aldhafeeri

**Affiliations:** National Centre for Environmental Technology (NCET), Life Science & Environment Research Institute (LSERI), King Abdulaziz City for Science & Technology (KACST), P.O. Box 6086, Riyadh 11442, Saudi Arabia; balharbi@kacst.edu.sa (B.H.A.); halhazmi@kacst.edu.sa (H.A.A.)

**Keywords:** air pollutants, COVID-19, lockdown, nitrogen oxides, ozone, AQI, air quality, Riyadh

## Abstract

This study investigated the concentrations of air pollutants (NO, NO_2_, NO_x_, SO_2_, CO, O_3_, PM_10_, and PM_2.5_) at three sites with different traffic loads (work, residential, and traffic sites) before, during, and after the COVID-19 lockdown. The main objective of this study was to evaluate the effects and associated potential pollution control implications of the lockdown on the quality of ambient air at three selected sites in the urban area of Riyadh City. The average concentrations of NO, NO_2_, NO_x_, and CO decreased during the lockdown period by 73%, 44%, 53%, and 32% at the work site; 222%, 85%, 100%, and 60% at the residential site; and 133%, 60%, 101%, and 103% at the traffic site relative to the pre-lockdown period, respectively. The average concentration of O_3_ increased by 6% at the work site, whereas the concentration of SO_2_ increased by 27% at the residential site and decreased by 6.5% at the work site. The changes in PM_10_ and PM_2.5_ varied and did not exhibit a clear pattern. The air quality index (AQI) results indicated that the contribution to “undesired” air quality by O_3_ was 35.29% of the lockdown period at the work site while contributions to undesired air quality by PM_10_ and PM_2.5_ were 75.6% and 100% at the work site, 94.5% and 100% at the residential site, and 96.7% and 100% at the traffic site, respectively. The findings of this study are useful for devising effective urban pollution abatement policies. Applying control measures comparable to the lockdown measures over one week will result in a decrease of approximately 19% and 15% in CO mean concentration and 25% and 18% in NO_2_ mean concentration at residential and traffic sites, respectively.

## 1. Introduction

In most of the world, including America, Europe, Oceania, Asia, and Africa, SARS-CoV-2 has continuously spread since 31 December 2019 [1,2]. However, it was only on 12 March 2020, that the World Health Organization (WHO) characterized COVID-19 as a pandemic and it had affected most of the world by then [3]. The high transmissibility and associated mortality rate of the virus [4] resulted in countries opting for different measures to contain it. These included a ban on public events, temporary shutting of all primary to higher academic institutions, encouragement of social distancing, near-total lockdowns, closure of non-essential businesses, and considerable reduction in public modes of transport such as buses, trains, and air travel. Although day-to-day human life was severely affected by these measures, the effect on air quality was generally positive [5,6,7,8,9]. 

Saudi Arabia (KSA) reported its first COVID-19 case on 2 March 2020. Thereafter, various measures were implemented to contain the pandemic. Suspension of the Umrah pilgrimage on March 4 was the first of these steps. On 8 March, all school and university activities were suspended. The third measure was implemented on 9 March, with a suspension of all international flights. Subsequently, on 23 March, a nationwide total lockdown from 07:00 to 18:00 (local time) was imposed by the Saudi government owing to the continuous increase in the number of COVID-19 cases and the non-availability of a treatment or vaccine and any other effective treatment worldwide. The subsequent measure was the limiting or rather stopping of movement between different regions of the country from March 25, resulting in a nationwide full lockdown on 6 April. On 28 May, except for Mecca, the movement between provinces was partially permitted, and the partial lifting of the lockdown began in all other cities, which included the opening of shopping malls. The subsequent relaxations were implemented on 31 May, for which, except for the Great Mosque of Mecca, prayers were permitted in all other mosques, and restrictions on restaurants, cafés, parks, and domestic flights were eased. On 21 June, the final lifting of the lockdown was implemented for all regions [10]. A total of 262,772 confirmed cases of COVID-19 and 2672 deaths were registered in Saudi Arabia from 2 March to 25 July 2020. In the capital of Riyadh, where our study was conducted, a partial curfew was enforced on the evening of 23 March 2020, from 19:00 to 06:00 every day for 21 days. On March 26, this curfew was increased to 15:00 to 06:00, with very limited exceptions involving life and safety until 14 April 2020. A 24 h curfew and total lockdown were imposed in Riyadh and all cities and regions across KSA for a four-day period, which coincided with the Eid holidays from 23 May to 27 May 2020. The 24 h curfew was replaced with a 15:00-to-06:00 curfew until 21 June, and a nationwide curfew was ended by the Saudi Arabian government. In addition, the restrictions on businesses were lifted after three months of lockdown.

Motor vehicles are a significant source of urban air pollution [11,12,13]. Transportation activities and use of vehicle transit were significantly reduced owing to the lockdown imposed as a proper protective measure to control and reduce the spread of COVID-19; this decrease in traffic was expected to significantly affect air pollution and air quality. For instance, the effect of partial lockdowns on the air quality in Rio de Janeiro, Brazil, was examined by Dantas et al. [5]; concentrations of NO_2_, CO, and PM_2.5_ decreased, while that of O_3_ increased. Sicard et al. [14] quantified the effect of the lockdown on air pollution in Wuhan and four other European cities, focusing on O_3_, PM_2.5_, PM_10_, and NO_x_, from 2017 to 2019 and the early months of 2020; a significant decrease was observed in PM_2.5_ and NO_2_ concentrations during the lockdown period, while O_3_ increased. The decrease in air pollutants reflected the decreased activities in road traffic and industrial and energy sectors. Concentrations of O_3_ were observed to be inversely proportional to those of NO_2_, with the former increasing as the latter decreased during lockdowns. Similar results have been reported in other countries. Sharma et al. [15] studied air pollution concentrations in 22 cities across India from 16 March to 14 April 2020, and compared the results with those of previous years up to 2017. They observed that the air pollution concentration varied by region and pollutant. However, overall, PM_2.5_ concentrations decreased in many regions that implemented lockdowns, while O_3_ concentrations increased. They concluded that air quality could be improved if regulatory authorities implemented stricter regulations [15]. Furthermore, Kanniah et al. [16] investigated spatial and temporal variations in different air pollutants and the aerosol optical depth (AOD) over Southeast Asian (SEA) countries, including Japan, Malaysia, Brunei, Singapore, and the Philippines. They observed up to a 70% decrease in AOD in the urban and industrial areas of Malaysia during the lockdown period (March to April 2020) compared with 2018 and 2019; however, in northern SEA countries, AOD values remained high even during the lockdown period owing to agricultural activities.

The national environmental strategy (NES) has been lunched by the government of Saudi Arabia in order to achieve the Saudi vision 2030. One of the priority areas of NES is “Global Warming and Air Pollution”. On the area of Global warming and air pollution, NES focuses on five topics one of them is “Reduction of automobile exhaust emissions”. Currently, strategies related to improving the situation of traffic such as streetlights control systems, vehicle weight and size restrictions, one-way streets, and road closures are implemented to improve air quality in the city. Additional measures that contribute to improving air quality include compliance with exhaust emission standards, banning the import of vehicles older than five years, emissions inventory development, and implementation of dispersion and receptor modeling. This study evaluated the expected subsequent change in the concentrations of air pollutants due to the decrease in traffic activities. This study explored the extent of variation in the concentrations of air pollutants and the associated air quality change during the lockdown in three different traffic-influenced environments (a low-traffic work site, moderate-traffic residential site, and heavy-traffic highway site). This could provide useful insights into and a better understanding of implementing proper regulatory plans to control and improve ambient air quality in the three types of investigated environments.

## 2. Materials and Methods

### 2.1. Study Area and Site Description

Riyadh City, the capital of Saudi Arabia, is the largest metropolitan area in the Arabian Peninsula, with an urban area of 1798 km^2^ inhabited by over 6.5 million people, most of whom depend on private means of transportation as well as taxi and car rental services for commuting. The air-pollution-monitoring sites in this study were selected to reflect the effects of the lockdown on work, residential, and traffic-influenced environments featuring low, moderate, and heavy levels of traffic emissions, respectively. Figure 1 shows a satellite image of Riyadh City and the air quality monitoring sites investigated in this study. The King Abdul Aziz City for Science and Technology (KACST) mobile air quality station (K-station) was located on the premises of KACST, which is a low-traffic environment. Moreover, the Almoroj air quality station (M-station) was located in a residential area with a moderate-traffic environment, whereas the air quality station of the King Fahad highway (F-station) was located in a heavy-traffic environment. The air quality stations of the Almoroj and King Fahad highway are approximately 5 km away from the KACST mobile air quality station and approximately 2.5 km from each other.

### 2.2. Air Pollution Measurement

The concentrations of air pollutants, including nitric oxide (NO), nitrogen dioxide (NO_2_), nitrogen oxides (NO_x_), sulfur dioxide (SO_2_), carbon monoxide (CO), ozone (O_3_), particulate matter equal to or less than 10 microns in diameter (PM_10_), and particulate matter equal to or less than 2.5 microns in diameter (PM_2.5_) were investigated in this study. The analysis provided in this paper is based on three measurement periods: April 2020 to June 2020 for the KACST mobile air quality station, and March 2020 to June 2020 and March 2019 to June 2019 for both air quality stations on the Almoroj and King Fahad highway. Over these study periods, continuous concentration measurements of seven air pollutants (NO, NO_2_, NO_x_, SO_2_, CO, PM_10_, and PM_2.5_) were obtained from air quality stations located in Almoroj and six air pollutants (NO, NO_2_, NO_x_, CO, PM_10_, and PM_2.5_) from King Fahad highway areas. These two air quality stations are operated by the Royal Commission for Riyadh. For the KACST mobile air quality station, eight air pollutants (O_3_, NO, NO_2_, NO_x_, SO_2,_ CO, PM_10_, and PM_2.5_) were measured.

In both air quality stations of Almoroj (M-station) and King Fahad highway (F-station), air pollution was measured using Environment SA analyzers. NO and NO_2_ were measured based on the chemiluminescence technology, the standard method for measuring nitrogen oxides (EN 14211), using a nitrogen oxide analyzer (Environment SA AC 32M) with the lowest detectable limit of <0.2 ppb. O_3_ was measured using an LED-based ultraviolet photometric O_3_ analyzer (Environment SA O_3_ 42M) with the lowest detectable limit of 0.2 ppb. SO_2_ was measured using ultraviolet fluorescence (UVF), the standard method of measuring SO_2_ (EN 14212), using an SO_2_ analyzer (Environment SA AF 22M) with the lowest detectable limit of <0.4 ppb. CO was measured using a non-dispersive infrared (NDIR) CO analyzer (Environment SA CO 12M) with the lowest detectable limit of 0.05 ppm. PM_10_ and PM_2.5_ were measured using a suspended particulate monitor (Environment SA MP101M) with the lowest detectable limit of 0.5 µg/m^3^ based on the standard ISO 10,473 beta gauge measurement method for the continuous measurement of concentration of fine dust in ambient air.

In the KACST mobile air quality station (K-station), air pollution was measured using HORIBA analyzers. NO and NO_2_ were measured based on a combination of the dual cross-flow modulation-type chemiluminescence principle and referential calculation method using a nitrogen oxide analyzer (HORIBA APNA-370), with the lowest detectable limit of 0.5 ppb. O_3_ was measured based on the non-dispersive ultraviolet absorption method (NDUV) in conjunction with the comparative calculation method using an O_3_ analyzer (HORIBA APOA-370), with the lowest detectable limit of 0.5 ppb. SO_2_ was measured based on the UVF using an SO_2_ analyzer (HORIBA APSA-370), with the lowest detectable limit of 0.5 ppb. CO was measured using an NDIR CO analyzer (HORIBA APMA-370) with the lowest detectable limit of 0.02 ppm. Non-methane hydrocarbon (NMHC) concentrations were measured using flame ionization detection (FID) with a selective combustion analyzer (HORIBA APHA-370), with the lowest detectable limit of 0.022 ppmC. PM_10_ and PM_2.5_ were measured using a Grimm EDM 365 dust monitor (Grimm Aerosol Technik GmbH, Ainring, Germany), with a resolution of 0.1 µg/m^3^.

### 2.3. Degree of Similarity

The degree of similarity or discrepancy of the air pollutants among the three air quality stations was calculated using the following convergence–divergence ratio (CD) [17]:$$CD_{jk}=\sqrt{\frac{1}{p}\sum_{i=1}^{p}{\left(\frac{xij-xik}{xij+xik}\right)}^{2}}$$
here, *xij* is the average concentration of pollutant *i* at a certain air quality station, *j* and *k* are two air quality stations, and *p* is the number of values representing each pollutant; three values (average, maximum, and minimum) were used to represent each pollutant. If the calculated CD tended towards zero, measurements from both air quality stations were considered to be similar, whereas if the CD was closer to one, measurements from the two air quality stations were considered to be different.

### 2.4. Air Quality Index

The air quality index (AQI) is a tool that assesses and describes the status of air quality and associated potential health implications. For evaluation and comparison, the AQIs for the three investigated sites were calculated according to the standard formulae of the United States Environmental Protection Agency (USEPA) and USEPA air quality standard limits. The measured concentrations of air pollutants were averaged to match the standard limits and classified according to the AQI breakpoints. In this study, the average times selected for AQI calculations for PM_10_, PM_2.5_, O_3_, NO_2_, CO, and SO_2_ were 24, 24, 1 and 8, 1, 1, and 1 h, respectively, and the breakpoints for each selected pollutant were according to the USEPA [18] indexing procedure (Appendix A). Typically, the scale of the AQI is divided into six general categories that are associated with health messages that convey the health implications of air quality and pollutant-specific health effects and sensitive groups. These categories are “Good,” “Moderate,” “Unhealthy for sensitive groups,” “Unhealthy,” “Very unhealthy,” and “Hazardous.” In this study, air quality with no observable health effects on humans was considered good whereas air quality in the Moderate, Unhealthy for sensitive groups, Unhealthy, very unhealthy, and Hazardous ranges that affect human health was considered as undesired.

## 3. Results and Discussion

### 3.1. Meteorology Measurements

Appendix A show the variation in the daily mean of air temperature, relative humidity, pressure, wind speed, and wind direction observed in the study area throughout the 2019 and 2020 study periods. The daily mean air temperature and relative humidity varied from approximately 14–40 °C and 6–61% in 2019 and from about 18–39 °C and 6–64% in 2020, respectively. The daily mean wind speed varied from 1.18–4.78 m/s in 2019 and from 1.15–4.56 m/s in 2020. The prevailing directions of airflow were southeasterly (~18% in 2019 and ~16% in 2020) followed by north-northeasterly (~10% in 2019 and ~13% in 2020), with wind speed predominantly occurring in the 1.38–3.06 m/s category (Appendix A).

### 3.2. Evaluation and Comparison of Air Pollutants Concentrations

A comparison of the air pollutant concentrations at the investigated sites and times indicated their differences among the residential, traffic, and work sites during the selected study periods. The CD method of comparison was applied to the concentrations of air pollutants, and the resultant values described the degree of similarity between the two sites. Similar sites had CD values approaching zero, whereas different sites had CD values approaching one. Figure 2 and Appendix A show the degree of similarity or discrepancy in the air pollutant concentrations among the selected residential, traffic, and work sites during the selected study periods. Generally, the highest dissimilarity was observed for NO_x_ with a CD value of 0.66, followed by CO and SO_2_ with CD values of 0.61 and 0.60, respectively. In contrast, low discrepancy or high similarity was observed for PM_10_ and PM_2.5_ concentrations during all investigated periods with CD values ranging from as low as 0.04 and up to 0.34 (Figure 2 and Appendix A).

In a pairwise comparison of the unrestricted periods, the highest dissimilarity between residential and traffic sites was indicated by a divergence value of 0.43 for NO, whereas the highest dissimilarity between residential and work sites was indicated by divergence value of 0.60 for SO_2_, and that between traffic and work sites was indicated by divergence value of 0.66 for NO_x_ (Figure 2 and Appendix A). In contrast, the least discrepancy, and thus the highest similarity, was observed between residential and work sites (divergence value of 0.12) for NO, followed by that between residential and traffic sites (divergence ratio of 0.17) for NO_2_ and that between residential and traffic sites (divergence value of 0.21) for CO (Figure 2 and Appendix A). The discrepancy in CO and NO_2_ pollutants between traffic and work sites (divergence ratios of 0.56 and 0.57, respectively) and between residential and work sites (divergence values of 0.40 and 0.56, respectively) exceeded the discrepancy in CO and NO_2_ pollutants between residential and traffic sites (divergence values of 0.21 and 0.17, respectively) (Figure 2 and Appendix A). Therefore, in terms of CO and NO_2_ concentrations during the unrestricted periods, sites more similar and dissimilar to the residential site (M-station) were the traffic site (F-station) and work site (K-station), respectively.

In a pairwise comparison of the lockdown period, the highest dissimilarity between residential and traffic sites was indicated by a divergence value of 0.46 for NO, while the highest dissimilarity between residential and work sites was indicated by a divergence value of 0.57 for SO_2_, and that between traffic and work sites was indicated by a divergence value of 0.61 for CO (Figure 2 and Appendix A). In contrast, the least discrepancy, and thus the highest similarity, was observed between the residential and work sites (divergence ratio of 0.04) for PM_2.5_, followed by that between the traffic and work sites (divergence value of 0.09) for PM_2.5_ and that between the residential and traffic sites (divergence value of 0.20) for CO (Figure 2 and Appendix A). The discrepancy in CO and NO_2_ pollutants between the traffic and work sites (divergence values of 0.61 and 0.46, respectively) and between the residential and work sites (divergence values of 0.50 and 0.53, respectively) exceeded the discrepancy in CO and NO_2_ pollutants between the residential and traffic sites (divergence values of 0.20 and 0.33, respectively) (Figure 2 and Appendix A). Therefore, in terms of CO and NO_2_ concentrations during the lockdown period, sites more similar and dissimilar to the residential site (M-station) were the traffic site (F-station) and work site (K-station), respectively. Moreover, Figure 2 and Appendix A also indicate that the lockdown decreased the range of CD values describing the similarity and discrepancy degrees among the three sites in terms of NO, NO_2_, and NO_x_, while the lockdown increased the range of CD values in terms of CO. The range of CD values in terms of NO, NO_2_, and NO_x_ decreased by 185% (from 0.12–0.49 to 0.36–0.49), 100% (from 0.17–0.57 to 0.33–0.53), and 207% (from 0.23–0.66 to 0.39–0.53), respectively. In contrast, the range of CD values in terms of CO increased by 15% (from 0.21–0.56 to 0.20–0.61).

### 3.3. Comparison of the Selected Periods

To evaluate the effect of the lockdown on air quality, the measured concentrations of air pollutants for two of the three studied sites (residential M-station and traffic F-station) during the lockdown (denoted by M-20 and F-20 for the residential and traffic sites, respectively) were compared with those recorded during two selected periods. The first period of these was the corresponding period of the lockdown in 2019 (denoted by M-19 and F-19 for the residential and traffic sites, respectively), and the second period was the 22 days before the lockdown in 2020 (denoted by M-pre-20 and F-pre-20 for the residential and traffic sites, respectively). For the work site (K-station), the measured concentrations of air pollutants during the lockdown (K-20) were compared only with those measured during the 22 days after the lockdown in 2020 (K-post) because of the unavailability of recorded data in 2019 and pre-lockdown in 2020. Figure 3 depicts a boxplot comparison of the hourly concentrations of air pollutants, including CO, NO, NO_2_, NO_x_, SO_2_, O_3_, and daily concentrations of particulate matter (PM_2.5_ and PM_10_). Generally, the interquartile ranges of most investigated gaseous air pollutants were wider for the traffic site than for the residential and work sites. All sites experienced significant decreases in the concentration levels of gaseous air pollutants during the lockdown period, except for SO_2_ at the residential site and O_3_ at the work site, both of which increased (Figure 3 and Figure 4). An increase in O_3_ concentrations was observed during the lockdown in many countries around the world [19,20,21,22,23]. In the three studied sites, the lockdown period compared with other periods also had the lowest median and mean concentration values of the investigated gaseous air pollutants, except for SO_2_ in the residential site and O_3_ at the work site (Figure 3). Moreover, the maximum outlier values of gaseous air pollutants were higher at traffic sites than at residential and work sites but generally were approximately comparable for PM_10_ and PM_2.5_ at all three sites (Figure 3). The PM_10_ and PM_2.5_ concentration changes were predominantly controlled by the frequent dust storms affecting the city; thus, the lockdown had only a slight effect on their concentration levels.

The percent changes in the concentrations of air pollutants at each station during the lockdown period are shown in Figure 4. The greatest decrease during lockdown was observed in the NO concentrations at all three sites. The residential and traffic sites experienced the highest decrease in the concentration levels of NO, NO_x_, and NO_2_, while the work site had the lowest decreases in concentration levels of air pollutants. Furthermore, the residential and work sites exhibited similar decreasing profiles (NO > NO_x_ > NO_2_ > CO). Comparing pollutant concentrations of the lockdown period with those of the corresponding period in 2019 and to the pre-lockdown period in 2020, the traffic site had comparable decreases in concentration levels of gaseous air pollutants, while the residential site exhibited comparable decreases only in concentration levels of NO_x_ and NO_2_. At this residential site, NO and CO had different decreases in concentration levels whereas SO_2_ increased in concentration during the lockdown period relative to the pre-lockdown period in 2020 but not with respect to the period in 2019. This indicated that the residential site experienced specific activities in 2019 (presumably construction activities), resulting in higher ambient concentrations of SO_2_ than those recorded during the lockdown period. The observed increase in SO_2_ concentration at the residential site during the lockdown relative to those of the pre-lockdown period in 2020 might be due to increased activities of heavy-duty diesel engines associated with construction activities near the site. However, this increase in construction activities was less than that in 2019. However, comparing PM_10_ concentrations during the lockdown period with those during the corresponding period in 2019 at the traffic site revealed a decrease of ~22% in PM_10_ concentrations during the lockdown, while the same comparison at the residential site increased by ~2% in PM_10_ concentrations during the lockdown. Assuming similar PM_10_ concentrations at the two sites resulting from dust storms affecting the city, this probably indicated that the residential site had additional emission sources of PM_10_ other than dust storms. In addition to this increase in PM_10_ at the residential site, no increases in pollutant concentrations were observed at the residential and traffic sites when the pollutant concentrations of the lockdown period were compared with those of the corresponding period in 2019. At the work site, the observed O_3_ concentration increase during the lockdown was due to the observed declining NO_x_ concentration levels and evidence of a hydrocarbon-limited regime in Riyadh, as previously reported [24]. In such a regime, the O_3_ production rate is limited by the supply of hydrocarbons, and O_3_ concentrations increase with increasing hydrocarbons and decrease with increasing NO_x_ [25,26,27].

Figure 5 shows the diurnal distribution of average hourly O_3_ and NO_x_ concentrations and their correlations during and after the lockdown at the work site (K-station). A typical systematic pattern of diurnal O_3_ change is characterized by a daytime high and nighttime low. This pattern was observed only during the period after the lockdown ended. Both periods (during and after the lockdown) had typical daytime maxima. However, during the lockdown period, the minima were at daytime rather than nighttime (Figure 5a,b). Following the diurnal variation in solar radiation, the O_3_ concentration increased gradually after sunrise and reached its highest concentration of ∼67 ppb at 14:00 during the lockdown and ∼75 ppb at 10:00–12:00 during the period after the lockdown and gradually declined thereafter (Figure 5a,b). The O_3_ concentration decreased to its lowest value of ∼32 ppb at 7:00 (after sunrise) during the lockdown and to ∼22 ppb at 5:00 (before sunrise) during the period after the lockdown (Figure 5a,b). Moreover, the nighttime O_3_ concentration during the lockdown period (ranging from 34.1 to 47.3 ppb) was higher than the nighttime O_3_ concentration during the period after the lockdown (ranging from 22.4 to 34.6 ppb). In addition, Figure 5c,d shows the diurnal patterns of NO_x_ corresponding to the same two periods. The anticorrelation between O_3_ and NO_x_ (R^2^ = 0.59) during the period after the lockdown is clearly illustrated by comparing Figure 5b,d,f, whereas this anticorrelation did not exist (R^2^ = ~0) during the lockdown period (Figure 5a,c,e). Furthermore, these same figures show that the NO_x_ concentration during the lockdown period lay in the range of 18.8–30.2 ppb during daytime (~6:00–19:00) and in the range of 17.6–24.5 ppb during nighttime. In contrast, the NO_x_ concentration during the period after the lockdown lay in the range of 6.9–51 ppb during the daytime and in the range of 27.3–57.7 ppb during nighttime. The reactions of NO with O_3_ (NO + O_3_ → NO_2_ + O_2_) and NO_2_ with O_3_ (NO_2_ + O_3_ → NO_3_ + O_2_) control the nighttime O_3_ concentration [28,29]. Therefore, the relatively lower nighttime NO_x_ concentration during the lockdown period compared with those during the period after the lockdown indicated lower O_3_ depletion by NO_x_ and explained the relatively higher nighttime O_3_ concentration during the lockdown period. Finally, the distribution of the hourly average O_3_ concentrations observed during the two periods is shown in Figure 5g,h. In these figures, hourly O_3_ concentrations are placed into predetermined 20 ppb bins. A maximum frequency value at O_3_ concentrations of 40–60 ppb was observed during both periods. However, O_3_ concentrations during the lockdown period followed a distribution that resembled a normal distribution to an extent, while those during the period after the lockdown exhibited a skewed distribution with a peak to the left (i.e., at low values) and a tail to the right (i.e., at high values). This type of skewed distribution indicated that anthropogenic pollution, particularly road traffic emissions, had a significant effect on the O_3_ concentrations observed during the period after the lockdown.

### 3.4. Analysis of Exceedances and Air Quality Index for Individual Pollutants

For evaluation and comparison, we calculated air pollutant exceedances and AQIs. At the three studied sites, the air pollutant exceedances during the lockdown period were calculated based on the General Authority for Meteorology and Environmental Protection (GAMEP) and USEPA standards (Appendix A). The AQIs in this study were calculated using the USEPA standard formulae and air quality standard limits, as stated in the Materials and Methods section. Air quality data covering the corresponding lockdown period in 2019, the pre-lockdown period in 2020, and the lockdown and post-lockdown periods were used to calculate the index values. PM_10_ and PM_2.5_ were averaged daily, and O_3_ was averaged every 1 and 8 h to match the breakpoint. The AQIs for individual pollutants during the selected periods are listed in Table 1. No hourly exceedances for CO, NO_2_, and SO_2_; daily exceedances for SO_2_; and 8 h exceedances for CO were observed in all stations during the entire lockdown period (Table 2). Similarly, the AQI for CO and SO_2_ had 0% undesired air at all stations during the lockdown (Table 1). However, the AQI for hourly NO_2_ concentrations revealed 0.7%, 1.06%, and 0.61% undesired air quality at the residential, traffic, and work sites, respectively, during the lockdown (Table 1). In contrast, the AQI for hourly NO_2_ concentrations had 6.99% undesired air quality during the corresponding lockdown period in 2019 and 4.39% undesired air quality during the pre-lockdown period in 2020 at the residential site. Similarly, the AQI for hourly NO_2_ concentrations had 6.99% undesired air quality during the corresponding lockdown period in 2019 and 3.74% undesired air quality during the pre-lockdown period in 2020 at the traffic site, while the AQI for hourly NO_2_ concentrations had 1.83% undesired air quality during the post-lockdown period at the work site. The percentages of undesired air quality reflected a good improvement in air quality in terms of NO_2_ at their respective sites during the lockdown since the other investigated periods (the corresponding lockdown period in 2019, pre-lockdown period in 2020, and post-lockdown period) experienced higher percentages of undesired air quality at these sites (Table 1).

For O_3_, the 8 h O_3_ concentration exceeded GAMEP and USEPA standards by 78 and 113 times, respectively, at the work site; 1 h O_3_ concentrations at the work site exceeded both GAMEP and USEPA standards by two times. At the work site, the lockdown period was better in compliance with the 1 h O_3_ standard than with the 8 h O_3_ standard. The 8 h GAMEP standard and USEPA standard were exceeded 39 and 66.5 times, respectively, more often than the 1 h standard during the lockdown period (Table 2). For human health, the 8 h O_3_ standard provides better protection than the 1 h standard. According to hourly AQI, the lockdown period had less undesirable air (0.05%) than the K-post 20. In contrast, based on the 8 h AQI, K-post 20 had less undesirable air (32.56%) than during the lockdown period. Therefore, exposure times longer than 1 h were of concern during the lockdown period. As for PM_10_, 24 h exceedances occurred at all stations. These exceedances occurred 8 and 33 times at the residential site, 9 and 35 times at the traffic site, and 6 and 29 times at the work site for the GAMEP and USEPA standards, respectively. Similarly, the 24 h exceedances for PM_2.5_ occurred at all stations, 56 times at the residential site, 52 times at the traffic site, and 51 times at the work site for the GAMEP and USEPA standards, respectively. Moreover, PM_2.5_ and PM_10_ had on average more than 75% of the measurements indicated as undesired air quality on the index (Table 1) and may have affected the health of the inhabitants of Riyadh City.

### 3.5. Insights for Improving Air Quality

The unintended restriction due to the lockdown could represent an opportunity to better understand potential emission control regulations and strategies and their implications. The concentrations of gaseous criteria air pollutants for the selected weeks were compared to evaluate the extent of these implications. Figure 6 shows the changes in concentrations of gaseous criteria air pollutants (CO, SO_2_, NO_2_, and O_3_) in residential, traffic, and work sites during the last week of the pre-lockdown period (the week before lockdown was imposed), the first and last weeks of the lockdown period, and the first week after the lockdown. To estimate whether the effects of the lockdown on pollutant concentrations were significant, an unpaired *t*-test was used to calculate the pollutant concentrations in the week before the lockdown and the first week of the lockdown (Table 3). In the traffic site (F-station), the median and mean CO concentrations during the first week of the lockdown period were less than those during the week before the lockdown. The CO mean and median concentrations of the pre-lockdown period decreased by 15% and 36%, respectively, in the first week of the lockdown period at the traffic site. However, this decrease was not significant at the 0.01 confidence level (Table 3). Moreover, the interquartile range was wider for the first week of the lockdown period than that for the week before the lockdown, reflecting a higher variability in the observed CO concentrations during the lockdown period. This higher variability in CO concentration was due to the low traffic during the lockdown hours and high traffic during hours exempted from the lockdown. The CO concentrations during the first week after the lockdown increased compared with those during the lockdown at the traffic site but did not revert to the levels of the week before imposing the lockdown. This increase was not significant at the 0.01 confidence level (Table 3).

The CO concentrations during the first week after the lockdown increased compared with those during the lockdown at the work site. This distinct upward change was significant at the 0.01 confidence level (Table 3). For the residential site (M-station), the CO concentrations exhibited similar trends of weak comparisons as those at the traffic site, except that the CO concentrations during the first week after the lockdown increased considerably compared with those during the lockdown and exceeded the levels of the week before the lockdown. Both the observed decrease in the CO concentrations in the first week of the lockdown period and the increase in the CO concentrations in the first week after the lockdown were significant at the 0.01 confidence level (Table 3). This suggested that the air quality benefit resulting from controlling CO emissions during this lockdown exhibited significant and more distinct changes in CO concentration levels at the residential and work sites than at the traffic site. Moreover, this change had a longer positive effect on air quality at the work and traffic sites than at the residential site.

NO_2_ and SO_2_ are directly emitted into the air from fuel combustion and industrial processes. NO_2_ concentrations in the traffic site (F-station) decreased by 15% and 18% in the median and mean, respectively, during the first week of the lockdown period and exhibited an interquartile range wider for the first week of the lockdown period than that for the week before the lockdown. The decrease in NO_2_ concentrations during the first week of the lockdown period was significant at the 0.01 confidence level (Table 3). Note that the NO_2_ concentrations exhibited a considerable increase during the week before the lockdown ended. During the lockdown period, construction activities were exempted from lockdowns. Consequently, the activities of heavy-duty diesel engines (bulldozers, dump and tanker trucks, compactors, cranes, diesel electrical generators, and road rollers) involved in the construction activities in the city of the Riyadh metro network, which has six lines and 85 stations, increased considerably with extended working hours, attaining a 24 h working mode during the last month of the lockdown period. When the lockdown ended, this considerable increase in construction activities returned to the normal pre-coronavirus operation level. In addition, electricity demand increased; thus, power generation in power plants increased due to progression in warmer conditions as the surface heating increased gradually during the March–May period. Therefore, the NO_2_ concentrations during the first week after the lockdown decreased only slightly compared with those during the week before the lockdown at the traffic site. This decrease was not significant at the 0.01 confidence level (Table 3). For the work site (K-station), the NO_2_ concentrations during the first week after the lockdown increased slightly (not significant at the 0.01 confidence level) compared with those during the lockdown. At the residential site (M-station), the NO_2_ concentrations decreased significantly during the first week after the lockdown was imposed compared with those during the week before the lockdown and an increase after the lockdown compared with those during the week before the lockdown. The 50th percentile of NO_2_ concentrations during the first week of the lockdown period was less than the lower quartile of the NO_2_ concentrations during the week before the lockdown. This indicated a significant decrease in the NO_2_ concentrations of more than two quarters between the pre-lockdown period and first week of the lockdown period in the residential site. This observed distinct decrease in the NO_2_ concentrations in the first week of the lockdown period was significant at the 0.01 confidence level, whereas the increase in the NO_2_ concentrations in the first week after the lockdown was not significant at the 0.01 confidence level (Table 3).

For SO_2_, during the first week after the lockdown, the concentration levels decreased only slightly at the residential site (M-station) compared with those during the week before the lockdown. This trend was understandable considering the exemption of construction activities from the lockdown and the gradual increase in power generation during the March–May period. Both the residential site (M-station) and work site (K-station) exhibited significant decreases in SO_2_ concentration levels after the lockdown compared with levels observed during the week before the lockdown (Figure 6 and Table 3).

For O_3_, comparing the last week of the lockdown period with the first week after the lockdown aided in assessing the persistence of the negative effect of the lockdown on elevated O_3_ concentration levels. Figure 6 shows that the O_3_ concentrations during the first week after the lockdown decreased only slightly (9.6% and 4.6% decrease in the mean and median, respectively) compared with those during the week before (the last week of the lockdown period).

Overall, the unintended experimental conditions provided by the COVID-19 lockdown provided valuable insights for improving air quality. Table 3 suggests that applying comparable control measures over one week will result in a decrease of approximately 19% and 15% in the mean CO concentration level at residential and traffic sites, respectively. Similarly, 25% and 18% reduction in the NO_2_ mean concentration level can be achieved at residential and traffic sites, respectively. Moreover, the emission control strategies equivalent to the measures implemented during the lockdown over one week could positively affect air quality in terms of controlling CO concentration levels that could last for approximately one week at work and traffic sites and a relatively shorter time at residential sites. Similarly, air quality benefits in terms of a decrease in NO_2_ concentration levels over one week could last for approximately one week at work sites and for a relatively shorter time at traffic and residential sites. However, caution should be applied when reducing NO_2_ concentration levels because it could result in an increase in O_3_ concentrations that could last for over a week in hydrocarbon-limited areas, particularly at work sites (Figure 6). In addition, strict inspection tests and rigorous standards for the emission compliance and working hours of trucks should be strategized because they can counteract any measures to improve air quality in terms of SO_2_ reduction.

## 4. Conclusions

The results of this study indicate that CO and NO_2_ concentrations at the residential and traffic sites were more similar than those at the work site during both the unrestricted and lockdown periods. The concentrations of these two gaseous criteria air pollutants (CO and NO_2_) declined, while the mean concentration of O_3_ increased at the work site and SO_2_ mean concentrations at the residential and work sites increased and decreased, respectively, during the lockdown period. The air quality improved significantly in terms of CO (reduction of 65% on average) and NO_2_ (reduction of 63% on average) concentrations. However, these improvements were counteracted by increased concentrations of O_3_ (increase of 5.8% on average) and SO_2_ (increase of 27% on average in residential sites). Air quality index (AQI) results indicated that the contribution to “undesired” air quality by O_3_ was 35.29% of the lockdown period at the work site. Efforts and efficient strategies to mitigate air pollution are required to ensure good air quality. This paper indicates that emission control strategies equivalent to the measures implemented during the lockdown over one week could positively affect air quality in terms of controlling CO concentration levels that could last for approximately one week at work and traffic sites and a relatively shorter time at residential sites. Similarly, air quality benefits in terms of a decrease in NO_2_ concentration levels over one week could last for approximately one week at work sites and a relatively shorter time at traffic and residential sites. In contrast, caution should be applied when reducing NO_2_ concentration levels because it could result in an increase in O_3_ concentrations that could last for over a week in hydrocarbon-limited areas.

Our results support decision-making and regulatory authorities in implementing effective regulatory plans to improve air quality. The results of this study could serve as insights for mitigating air pollution levels and might aid policymakers in revising the existing policies and strategies for controlling air pollution and subsequently help in improving air quality for a healthy sustainable environment.

## Figures and Tables

**Figure 1 ijerph-19-00727-f001:**
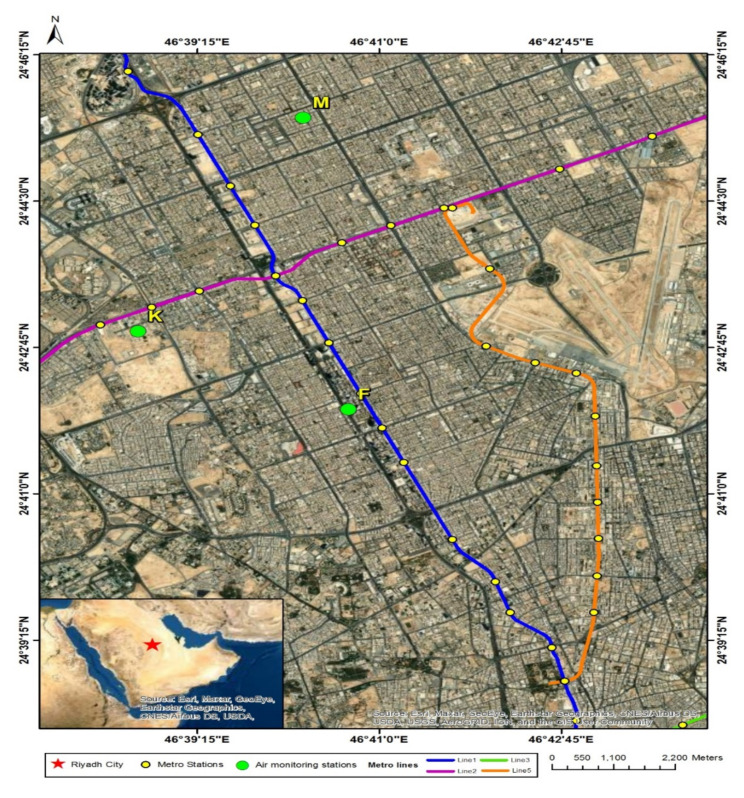
Locations of air-quality-monitoring stations used in this study and metro stations in Riyadh City.

**Figure 2 ijerph-19-00727-f002:**
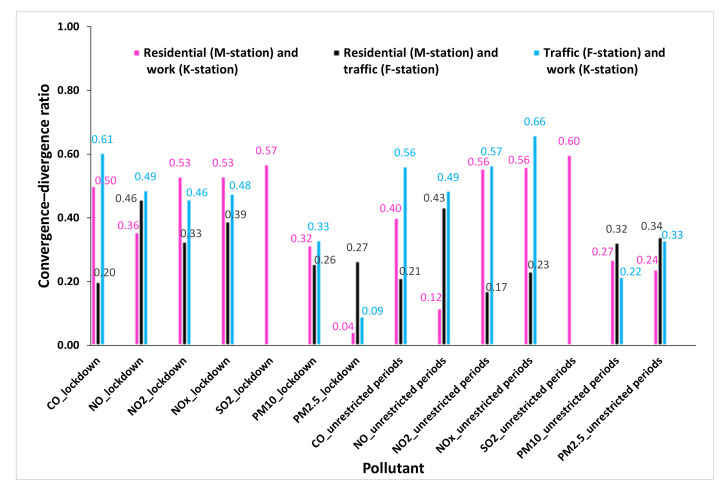
The degree of similarity or discrepancy in the air pollutant concentrations among the selected residential, traffic, and work sites during the selected study periods.

**Figure 3 ijerph-19-00727-f003:**
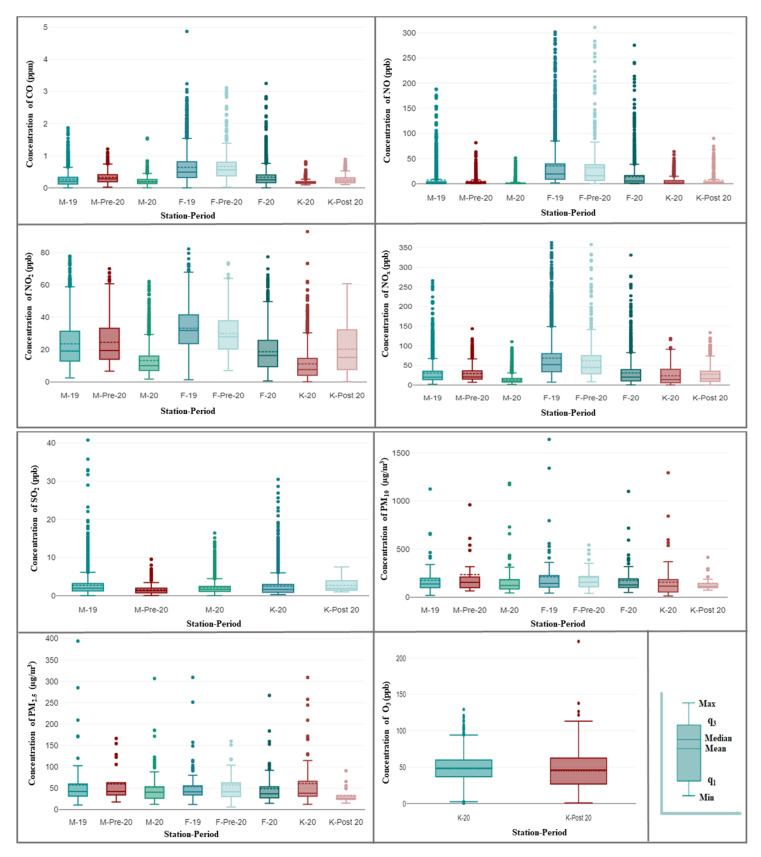
Boxplots of the distributions of 1 h CO, 1 h NO, 1 h NO_2_, 1 h NO_x_, 1 h SO_2_, 1 h O_3_, 24 h PM_2.5_, and 24 h PM_10_ during different periods at three air quality stations (M, F, and K).

**Figure 4 ijerph-19-00727-f004:**
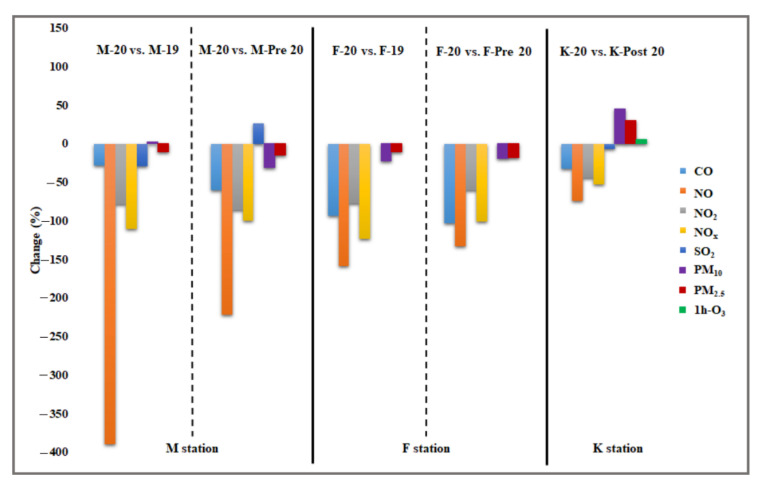
Comparison of air pollutants during the lockdown period for the three stations with the corresponding period in 2019 (M and F stations), pre-lockdown period (M and F stations), and post-lockdown period (K station).

**Figure 5 ijerph-19-00727-f005:**
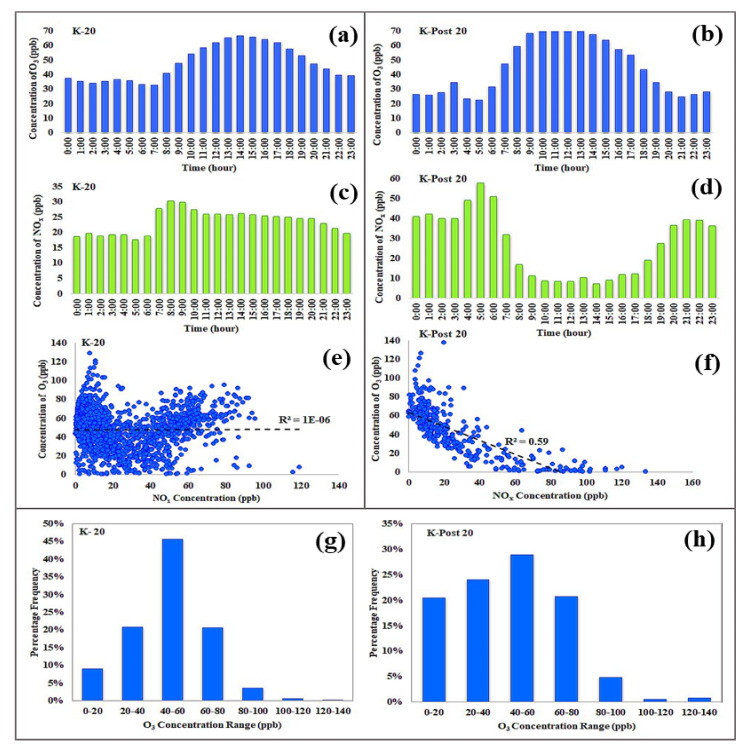
Diurnal distribution of average hourly O_3_ (**a**,**b**) and NO_x_ concentrations (**c**,**d**), linear correlations between O_3_ and NO_x_ (**e**,**f**), and frequency distribution of hourly O_3_ (**g**,**h**) during (K-20) and after the lockdown period (K-Post 20) at the work site (K station).

**Figure 6 ijerph-19-00727-f006:**
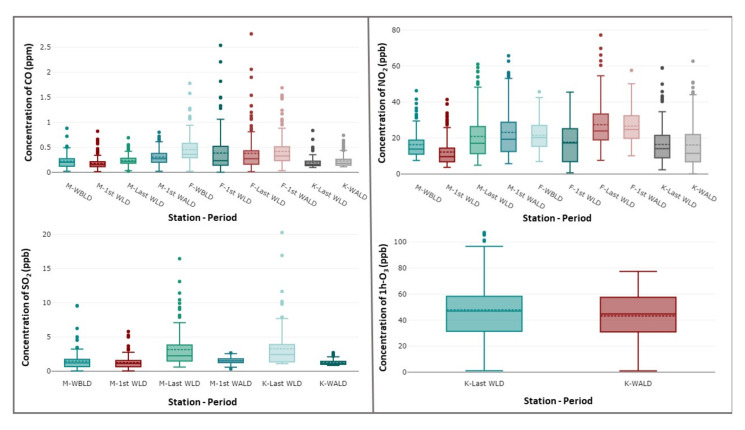
Concentrations of gaseous criteria air pollutants (CO, SO_2_, NO_2_, and O_3_) in the residential, traffic, and work sites during the last week of pre-lockdown period (the week before the lockdown), the first and last week of lockdown period, and the first week after the lockdown.

**Table 1 ijerph-19-00727-t001:** Percentage of undesirable air (moderate, unhealthy for sensitive groups, unhealthy, very unhealthy, and hazardous) as indicated by AQIs for individual pollutants during the corresponding lockdown period in 2019, pre-lockdown period in 2020, lockdown, and post-lockdown periods.

Station-Period	CO	SO_2_	NO_2_	O_3_		PM_2.5_	PM_10_
	1 h	1 h	1 h	1 h	8 h	24 h	24 h
**M-19**	0% Und.	0.09% Und.	6.99% Und.	-	-	98.99% Und.	96.70% Und.
**M-Pre 20**	0% Und.	0% Und.	4.39% Und.	-	-	100% Und.	100% Und.
**M-20**	0% Und.	0% Und.	0.70% Und.	-	-	100% Und.	94.51% Und.
**F-19**	0% Und.	-	6.99% Und.	-	-	100% Und.	93.40% Und.
**F-Pre 20**	0% Und.	-	3.74% Und.	-	-	95.50% Und.	95.50% Und.
**F-20**	0% Und.	-	1.06% Und.	-	-	100% Und.	96.70% Und.
**K-20**	0% Und.	0% Und.	0.61% Und.	0.05% Und.	35.29% Und.	100% Und.	75.61% Und.
**K-Post 20**	0% Und.	0% Und.	1.83% Und.	0.38% Und.	32.56% Und.	100% Und.	100% Und.

Und.: Undesirable air (moderate, unhealthy for sensitive groups, unhealthy, very unhealthy, and hazardous).

**Table 2 ijerph-19-00727-t002:** Air pollutant exceedances with reference to GAMEP and USEPA standards during the lockdown period at the three studied sites (residential (M), traffic (F), and work (K)).

		Exceedances					
Averaging Time	Pollutant	M-Station		F-Station		K-Station	
		GAMEP	USEPA	GAMEP	USEPA	GAMEP	USEPA
1 h	NO_2_	0	0	0	0	0	0
	O_3_	-	-	-	-	2	2
	SO_2_	0	0	-	-	0	0
	CO	0	0	0	0	0	0
8 h	O_3_	0	0	-	-	78	133
	CO	0	0	0	0	0	0
24 h	PM_10_	8	33	9	35	6	29
	PM_2.5_	56	56	52	52	51	51
	SO_2_	0	0	0	0	0	0

**Table 3 ijerph-19-00727-t003:** Weekly comparisons of the gaseous criteria air pollutants measured at the different sites in Riyadh before and after the lockdown and relative difference between the mean values (↓ denotes a decrease, ↑ denotes an increase, and * denotes significant change at the 0.01 confidence level according to the *t*-test).

	WbLD–1st WLD			Last WLD–1st WALD		
	Sites					
	Residential	Traffic	Work	Residential	Traffic	Work
CO	19.2 ↓ *	15.1 ↓	-	28.3 ↑ *	10.1 ↑	18.0 ↑ *
NO_2_	25.1 ↓ *	17.6 ↓ *	-	10.8 ↑	2.9 ↓	1.8 ↓
SO_2_	14.0 ↓		-	51.4 ↓ *		60.7 ↓ *
O_3_			-			9.7 ↓

## Data Availability

The datasets used and/or analyzed during the current study are available from the corresponding author on reasonable request.

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
