# Peer review of "Air Quality of Work, Residential, and Traffic Areas during the COVID-19 Lockdown with Insights to Improve Air Quality"

_ijerph, 2022, doi:10.3390/ijerph19020727_

Round 1

Reviewer 1 Report

Introduction

The analysis of the lockdown measures related to the COVID-19 pandemic makes it possible to estimate the effectiveness of these measures on air quality. Those studies are interesting when preparing air quality improvement plans, where similar measures are usually used (although with less intensity).

The division into three different areas (work, residential areas and traffic sites) allows to analyze the differentiated behavior that pollutants present based on the different patterns of human behavior.

Major comments

  1. The results of the CD method are presented exclusively in a table within the section "non-published material". It should have a graph within the article to more appreciate the values obtained. It should be analyzed the type of graph that allows the results achieved to be clearly shown.
  2. It would make more sense to compare the three stations with similar periods. As it is indicated that the K station does not have data in the K-prev-20 period, it would probably have been more interesting to have compared the M and F stations in the same period (M-post-20 and F-post-20).
  3. The authors indicate the absence of data for station K in the period called K-prev-20 (“For the work site (K-station), the measured concentrations of air pollutants during the lockdown (K-20) were compared only with those measured during the 22 days after the lockdown in 2020 (K-post) because of the unavailability of recorded data in 2019 and pre-lockdown in 2020”). However, Figure 2 shows this period for some pollutants. It should be checked to verify that this is an error in the tag used, which should be “K-post-20”.

Minor comments

  1. Although it appears in section 2.2, the relationship between the name of the stations and the initial used in figure 1 should appear in section 2.1 or in the legend of said figure. In the first reading, the association that appears later is missing.
  2. In the first paragraph of section 3.1, the CD method used appears redundantly explained, which has already been explained immediately before. “The CD method of comparison was applied to the concentrations of air pollutants, and the resultant values described the degree of similarity between the two sites. Similar sites had CD values approaching zero, whereas different sites had CD values approaching one”. Formally, in the results section, the methods used should not be reiterated so explicitly, to simplify the wording.
  3. A new section (3.2) should be entered on line 260. At this point, the comparison via the CD method is completed and the comparison of periods begins.
  4. There are other types of graphs that allow representing the values of figure 4, showing not only the mean but also the standard deviation obtained in the period studied. It is advisable to represent the values using methods widely used in air quality, such as those contained in the openair library of R.
  5.  

Author Response

We are grateful for the editor and the reviewers for their valuable and constructive reviews, comments, and suggestions to improve this manuscript. Our responses are provided point-by-point for each Reviewer as follows.

Reviewer #1:

Reviewer #1(Introduction):

The analysis of the lockdown measures related to the COVID-19 pandemic makes it possible to estimate the effectiveness of these measures on air quality. Those studies are interesting when preparing air quality improvement plans, where similar measures are usually used (although with less intensity). The division into three different areas (work, residential areas and traffic sites) allows to analyze the differentiated behavior that pollutants present based on the different patterns of human behavior.

___________________________________________________________________________________

Reviewer #1 (Major comments):

  1. 1. The results of the CD method are presented exclusively in a table within the section "non-published material". It should have a graph within the article to more appreciate the values obtained. It should be analyzed the type of graph that allows the results achieved to be clearly shown.

Response 1:

We are grateful for the reviewer's comment. We have added a new figure as Figure 2.

Figure 2. The degree of similarity or discrepancy in the air pollutant concentrations among the selected residential, traffic, and work sites during the selected study periods.

  1. 2. It would make more sense to compare the three stations with similar periods. As it is indicated that the K station does not have data in the K-prev-20 period, it would probably have been more interesting to have compared the M and F stations in the same period (M-post-20 and F-post-20).

Response 2:

We thank the reviewer and appreciate the suggestion. However, we believe it would be more helpful to be consistent concerning the periods of comparison to avoid complexity and confusion. Moreover, the manuscript already has many comparisons and figures and it is long as stated by Reviewer #2 “Another snag is that the manuscript is very long.”.

  1. The authors indicate the absence of data for station K in the period called K-prev-20 (“For the work site (K-station), the measured concentrations of air pollutants during the lockdown (K-20) were compared only with those measured during the 22 days after the lockdown in 2020 (K-post) because of the unavailability of recorded data in 2019 and pre-lockdown in 2020”). However, Figure 2 shows this period for some pollutants. It should be checked to verify that this is an error in the tag used, which should be “K-post-20”.

Response 3:

Thank you for the reviewer, we have re-checked Figure 2 (Figure 3 in the manuscript revision) and corrected “K-pre 20” to be “K-post” in the Figure.

Reviewer #1(Minor comments):

  1. 1. Although it appears in section 2.2, the relationship between the name of the stations and the initial used in figure 1 should appear in section 2.1 or in the legend of said figure. In the first reading, the association that appears later is missing.

Response 4:

Thank you for the reviewer, we have added the initial of stations in section 2.1.

“The King Abdul Aziz City for Science and Technology (KACST) mobile air quality station (K-station) was located on the premises of KACST, which is a low-traffic environment. Moreover, the Almoroj air quality station (M-station) was located in a residential area with a moderate-traffic environment, whereas the air quality station of the King Fahad highway (F-station) was located in a heavy-traffic environment.”.

  1. 2. In the first paragraph of section 3.1, the CD method used appears redundantly explained, which has already been explained immediately before. “The CD method of comparison was applied to the concentrations of air pollutants, and the resultant values described the degree of similarity between the two sites. Similar sites had CD values approaching zero, whereas different sites had CD values approaching one”. Formally, in the results section, the methods used should not be reiterated so explicitly, to simplify the wording.

Response 5:

In response to the comment, we have removed the explanation of CD method from the results section.

  1. 3. A new section (3.2) should be entered on line 260. At this point, the comparison via the CD method is completed and the comparison of periods begins.

Response 6:

Thank you for the reviewer, we have added a new section “3.2 comparison the selected periods”.

3.2 Comparison of the selected periods

To evaluate the effect of the lockdown……”

  1. There are other types of graphs that allow representing the values of figure 4, showing not only the mean but also the standard deviation obtained in the period studied. It is advisable to represent the values using methods widely used in air quality, such as those contained in the openair library of R.

Response 7:

We thank the reviewer and appreciate the suggestion. We are not familiar with the openair library of R. However, we certainly will consider it in future work.

Reviewer 2 Report

The manuscript is an interesting study of air pollutant concentration during lockdown periods.

The overall quality of the paper is high. A good and complete introduction is present as well as the methodological description and the discussions of findings.

I have one major concern: I did not find a meteorological analysis of the comparing periods. Meteorological conditions are fundamental in order to understand if we are facing comparable situations. I miss this kind of information. I see they are utilizing hourly data for similar periods of 2019 and 2020 but, in my opinion, this is not sufficient to assume that changes in local transport did not occur.

I suggest the author improve the paper assuring the reader that their findings are not affected by different wind speeds and directions which can alter their conclusions.

Author Response

We are grateful for the editor and the reviewers for their valuable and constructive reviews, comments, and suggestions to improve this manuscript. Our responses are provided point-by-point for each Reviewer as follows.

Reviewer #2:

Reviewer #2 (Comments and Suggestions for Authors):

The manuscript is an interesting study of air pollutant concentration during lockdown periods. The overall quality of the paper is high. A good and complete introduction is present as well as the methodological description and the discussions of findings. I have one major concern: I did not find a meteorological analysis of the comparing periods. Meteorological conditions are fundamental in order to understand if we are facing comparable situations. I miss this kind of information. I see they are utilizing hourly data for similar periods of 2019 and 2020 but, in my opinion, this is not sufficient to assume that changes in local transport did not occur. I suggest the author improve the paper assuring the reader that their findings are not affected by different wind speeds and directions which can alter their conclusions.

Response 8:

We thank the reviewer for the supportive comments and valid concern. Unfortunately, the meteorological data is not available. Furthermore, we totally agree that wind speed and direction within an urban area vary from place to place. However, we expect that the change in wind speed and direction would be small and all three investigated sites would be under the same background meteorological conditions due to the fact that they are just around 2.5 to 5 km apart from each other as was stated in section 2.1.

Reviewer 3 Report

These authors used the natural experiment of the strict lockdown that took place last Spring 2020 in the capital city of Saudi Arabia to describe its effect on the quality of ambient air regarding the main gaseous particulate and pollutants.  In this context they wisely chose to compare between them different areas of the city characterized by different intensity of motor traffic but also of residential and construction activities, and also chose to compare these values of the main pollutants as recorded during the same period of the previous year at a time when there was no restriction to traffic and human activities in the capital city. Not surprisingly, and in agreement with similar findings obtained in other areas of the world during the pandemic-induced lockdown, the authors found a decrease of most pollutants with the exception of ozone, which in contrast was increased with an inverse relationship with the decrease of NO2 concentrations.

On the whole, this study is well done and is also well written, although in a way that is very technical and unlikely to be easily grasped by a non-specialized readership. Limitations of this manuscript are that the findings are not new and that the implications are mainly cogent for the country and city of the authors. Another snag is that the manuscript is very long, but I recognize that notwithstanding it is of easy perusal. At the end the authors offer an interesting suggestion based on their data, i.e., that relatively brief and reversible periods of strict lockdown would significantly benefit albeit transiently the air quality in areas characterized by high levels of air pollution.   

Author Response

We are grateful for the editor and the reviewers for their valuable and constructive reviews, comments, and suggestions to improve this manuscript. Our responses are provided point-by-point for each Reviewer as follows.

Reviewer #3:

Reviewer #3 (Comments and Suggestions for Authors):

These authors used the natural experiment of the strict lockdown that took place last Spring 2020 in the capital city of Saudi Arabia to describe its effect on the quality of ambient air regarding the main gaseous particulate and pollutants.  In this context they wisely chose to compare between them different areas of the city characterized by different intensity of motor traffic but also of residential and construction activities, and also chose to compare these values of the main pollutants as recorded during the same period of the previous year at a time when there was no restriction to traffic and human activities in the capital city. Not surprisingly, and in agreement with similar findings obtained in other areas of the world during the pandemic-induced lockdown, the authors found a decrease of most pollutants with the exception of ozone, which in contrast was increased with an inverse relationship with the decrease of NO2 concentrations.

On the whole, this study is well done and is also well written, although in a way that is very technical and unlikely to be easily grasped by a non-specialized readership. Limitations of this manuscript are that the findings are not new and that the implications are mainly cogent for the country and city of the authors. Another snag is that the manuscript is very long, but I recognize that notwithstanding it is of easy perusal. At the end the authors offer an interesting suggestion based on their data, i.e., that relatively brief and reversible periods of strict lockdown would significantly benefit albeit transiently the air quality in areas characterized by high levels of air pollution.

Response 9:

We thank the reviewer for the supportive comments. We have done our best to keep this manuscript as short as possible. However, we could not make it shorter than it is now due to the many aspects covered in this study.

Reviewer 4 Report

Please review the attached file.

Author Response

We are grateful for the editor and the reviewers for their valuable and constructive reviews, comments, and suggestions to improve this manuscript. Our responses are provided point-by-point for each Reviewer as follows.

Reviewer #4:  (Comments and Suggestions for Authors):

The rationale is well explained in the Introduction, references are correctly cited, data collection is logically designed, results are well presented and interpreted. I think that this article could be published in IJERPH. The abstract part of the study is written too long. Authors should keep this part short with more precise terms.

Response 10:

We thank the reviewer for the supportive comments. In order to shorten the abstract as requested by the reviewer, we have deleted the last sentence in the abstract (“Significant mitigation of urban air pollution could be accomplished through intermittent implementations of strict pollution control measures.”).

 Page 1 – Abstract line 25: ‘Significant mitigation of urban air pollution could be accomplished through intermittent implementations of strict pollution control measures.’ What do the authors want to emphasize in this sentence? Is it necessary to have an occasional shutdown to reduce air pollution?

Response 11:

We thank the reviewer for the comment. Yes, we believe that strict pollution control measures that would result in a situation comparable to that during the lockdown can be achieved in high traffic emission areas through implementation of occasional banning of hydrocarbon-based fuel vehicles and make these areas only accessible by pure electric means of private and public transportation.

Page 2 – Line 69: ‘Motor vehicles are a significant source of urban air pollution.’ This is a simple correct sentence, but this sentence needs a reference for proof.

Response 12:  As requested by the reviewer, we added some references.

“Motor vehicles are a significant source of urban air pollution (Faiz,1993; Mayer,1999; Ghose et al., 2004 ).”

  1. FAIZ, A. Automotive emissions in developing countries-relative implications for global warming, acidification and urban air quality. Transportation Research Part A: Policy and Practice, 1993, 27.3: 167-186.‏
  2. Ghose, M. K.; Paul, R.; Banerjee, S. K. Assessment of the impacts of vehicular emissions on urban air quality and its management in Indian context: the case of Kolkata (Calcutta). Environmental Science & Policy, 2004, 7.4: 345-351.‏
  3. Mayer, H. Air pollution in cities. Atmospheric environment, 1999, 33.24-25: 4029-4037.‏

Page 4 – Line 84: This equation should be edited as follows: ????=√1?Σ(???−??????+???)??=12

Response 13:

Thank you for the comment. In the manuscript, we used the formula that we are familiar to as it was stated by Wongphatarakul et al., 1998.

 Page 11 – Line 426: Table 1 should be corrected visually.

Response 14:

Thank you for the comment. As requested by the reviewer, Table 1 in the manuscript has been updated to be more visible.

Station-Period

CO

SO2

NO2

O3

PM2.5

PM10

1-h

1-h

1-h

1-h

8-h

24-h

24-h

M-19

0 % Und.

0.09 % Und.

6.99 % Und.

-

-

98.99 % Und.

96.70 % Und.

M-Pre 20

0 % Und.

0 % Und.

4.39 % Und.

-

-

100 % Und.

100 % Und.

M-20

0 % Und.

0 % Und.

0.70 % Und.

-

-

100 % Und.

94.51 % Und.

F-19

0 % Und.

-

6.99 % Und.

-

-

100 % Und.

93.40 % Und.

F-Pre 20

0 % Und.

-

3.74 % Und.

-

-

95.50 % Und.

95.50 % Und.

F-20

0 % Und.

-

1.06 % Und.

-

-

100 % Und.

96.70 % Und.

K-20

0 % Und.

0 % Und.

0.61 % Und.

0.05 % Und.

35.29 % Und.

100 % Und.

75.61 % Und.

K-Post 20

0 % Und.

0 % Und.

1.83 % Und.

0.38 % Und.

32.56 % Und.

100 % Und.

100 % Und.

Und.: Undesirable air (moderate, unhealthy for sensitive groups, unhealthy, very unhealthy, and hazardous).

Page 14 – Line 563: ….air pollution levels and might aid policymakers in revising the existing policies and strategies for controlling….What are the current policies and strategies? Information or example can be provided. Overall, the study was well handled by the authors. This article can be published if the authors edited the deficiencies mentioned above.

Response 15:

We thank the reviewer for the comment. In this revised manuscript, we have added information and example in page 2 and 3 line 94-103 as follows:

“The national environmental strategy (NES) has been lunched by the government of Saudi Arabia in order to achieve the Saudi vision 2030. One of the priority areas of NES is “Global Warming and Air Pollution”. On the area of Global warming and air pollution, NES focuses on five topics one of them is “Reduction of automobile exhaust emissions”. Currently, strategies related to improving the situation of traffic such as street lights con-trol systems, vehicle weight and size restrictions, one way streets, and road closures are implemented to improve air quality in the city. Additional measures that contribute to improving air quality include compliance with exhaust emission standards, banning the import of vehicles older than five years, emissions inventory development, and implementation of dispersion and receptor modeling.”.

Round 2

Reviewer 2 Report

I apologise with the authors: my concern about the meteorological conditions was not 'between' sites, I agree the small distance allows to consider small differences. I referred to the different periods of the experiment: " The analysis provided in this paper is based on three measurement periods: April 2020 to June 2020 for the KACST mobile air quality station, and March 2020 to June 2020 and March 2019 to June 2019 for both air quality stations..." the question is if in these different years and periods the authors can certainly assess the meteorological conditions were comparable. I'm sure it is possible to implement the paper with a meteorological analisys of March-June 2019 and April-June 2020 from the local meteorological network in order to say that are comparable. I did not propoperly expressed my concern, I'm sorry for the inconvenient.

Author Response

We thank the reviewer for the clarification. We have added additional table, figure, and paragraph as follows.

3.1 Meteorology measurements:

Figure 1S and Table 2S show the variation in the daily mean of air temperature, relative humidity, pressure, wind speed, and wind direction observed in the study area throughout the 2019 and 2020 study periods. The daily mean air temperature and relative humidity varied from approximately 14-40 °C and 6-61 % in 2019 and from about 18-39 °C and 6-64 % in 2020, respectively. The daily mean wind speed varied from 1.18-4.78 m/s in 2019 and from 1.15-4.56 m/s in 2020. The prevailing directions of airflow were southeasterly (~18% in 2019 and ~16% in 2020) followed by north-northeasterly (~10% in 2019 and ~13% in 2020), with wind speed predominantly occurring in the 1.38-3.06 m/s category (Table 2S).”
